# Melanotrichoblastic Carcinosarcoma: A Histopathological Case Report of a Previously Undescribed Nosological Unit

**DOI:** 10.3390/reports8040218

**Published:** 2025-10-29

**Authors:** George Stoyanov, Hristo Popov

**Affiliations:** 1Department of Pathology, Multiprofile Hospital for Active Treatment, Vasil Aprilov 61 Str., 9700 Shumen, Bulgaria; 2Department of General and Clinical Pathology, Forensic Medicine and Deontology, Faculty of Medicine, Medical University—Varna, Hr. Smirnensi 1 Blvd, 9000 Varna, Bulgaria

**Keywords:** trichoblastoma, melanotrichoblastoma, trichoblastic carcinoma, trichoblastic carcinosarcoma, melanotrichoblastic carcinosarcoma, melanotic trichoblastic carcinosarcoma

## Abstract

**Background and Clinical Significance**: Trichoblastomas and their variants are rare and underrecognized tumors, with their differential diagnosis being predominantly the much more common basal cell carcinoma. Variants of trichoblastoma, such as melanotrichoblastoma, and malignant counterparts, such as trichoblastic carcinoma and trichoblastic carcinosarcoma, are also rare and probably further underrecognized. **Case Presentation**: Herein, we present the morphological findings of a tumor located on the right arm of an 86-year-old female patient. The tumor presented with a mixed morphology comprising malignant epithelial nests and retiform structures with focal keratinization and comedo-type necrosis, admixed with dendritic melanocytes, and it had a strikingly bizarre and hypercellular stroma. Immunohistochemistry was positive for BerEp4 in the epithelial nest, HMB-45 was positive only in dendritic melanocytes admixed in some of the nests, and CK20 showed a focal dot-like reaction in intermixed Merkel cells. The stroma was negative for epithelial and melanocytic markers; however, it also exhibited hypercellularity and a bizarre appearance, in addition to a high Ki-67 proliferative index, which further defined it as malignant. Based on the morphology and immunohistochemical profile, the tumor was defined as melanotrichoblastic carcinosarcoma—a previously undescribed nosological unit with unique morphology. **Conclusions**: Melanotrichoblastomas, as well as trichoblastic carcinosarcomas, are exceedingly rare and underrecognized tumors due to their mimicry of other, more common malignancies. The currently presented case, to the best of the authors’ knowledge, is the first reported one of melanotrichoblastic carcinosarcoma.

## 1. Introduction and Clinical Significance

Cutaneous tumors with trichoblastic differentiation, designated as trichoblastomas, are rare and underrecognized entities [1,2]. This is primarily due to their close morphological appearance to a much more common cutaneous neoplastic process—basocellular carcinoma [1,3]. Both groups of tumors predominantly comprise dermally located basaloid cell nests with peripheral palisading [1,3]. Several distinct features are, however, present, with basal cell carcinoma having an epidermal or, in rare cases, hair shaft connection, and often mucin production with a cellular nest clefting towards the reactive derma [1,3,4].

Tumors with trichoblastic differentiation, on the other hand, lack the clefting and have a specialized secondary component comprising a cellular stroma wrapping the cellular nest, recreating the physiological structure of the hair shaft [4]. This stroma often intersects the basaloid nest and produces mesenchymal papillary bodies [2]. However, while basal cell carcinoma is indeed an exceedingly common malignant tumor, despite the distinct features described, trichoblastic tumors remain underrecognized and underreported [1,4]. Chief among these is their rarity, and secondly, in some cases, features may overlap. Due to fixation artifacts, there may be some clefting present in trichoblastomas; however, this is always associated with a lack of mucin production [2]. Secondly, reactive dermal changes in basal cell carcinoma may produce a hypercellular dermis that mimics the specialized stroma of trichoblastomas [2]. Lastly, even though basal cell carcinoma, as a malignant tumor, presents with at least minimal cell and nuclear atypia and increased mitotic activity, in cases of long-lasting trichoblastomas, a degenerative atypia may mimic that seen in basal cell carcinoma, and in larger and rapidly growing lesions, mitotic activity may be prevalent [2,4]. From a clinical standpoint, the diagnosis of trichoblastomas is rarely suspected, and a basal cell carcinoma diagnosis is not often questioned, as both tumors develop predominantly in the elderly and have a high affinity for the sun-exposed skin of the head and neck [2]. While in their biological nature, trichoblastomas and basal cell carcinoma differ significantly, with one being benign and the other, while having negligible metastatic potential, is still a locally aggressive malignant tumor, the definitive treatment for both is surgical excision.

Further underlining the rarity of trichoblastomas is the presence of rare and probably underidentified variants, which further support the notion that this benign tumor completely recapitulates the physiological structure of the hair follicle in a hamartomous manner [4]. The most commonly recognized variants are trichoblastomas and trichoepithelioma, which are included in the current revision of the World Health Organization. Classification of skin tumors recognizes the same condition. Apart from the two components already mentioned—basaloid epithelial and mesenchymal—these tumors often have a prominent Merkel cell population, which is often used as a diagnostic aid in the differential diagnosis with basal cell carcinoma, by means of focal cytokeratin 20 dot-like positivity [2,4].

Over the previous two and a half decades, a fourth component has also been reported, although not specifically recognized by the WHO, with the presence of dendritic melanocytes within the lesion, further recapitulating the hair follicle structure [4,5,6,7,8,9,10,11,12]. These exceedingly rare lesions, deemed melanotrichoblastoma, of which only eight have been published to date, further increase their differential diagnosis to include melanoma [4,5,6,7,8,9,10,11,12].

As already mentioned, trichoblastomas are benign in nature, but they do have two malignant counterparts, which probably develop from them. These encompass trichoblastic carcinoma and trichoblastic carcinosarcoma [2,13,14,15,16]. Trichoblastic carcinoma is the more common of the two conditions, in which only the basaloid epithelial cell component undergoes malignant transformation and makes the differential diagnosis with basal cell carcinoma even more difficult [2,14,17,18,19,20]. In trichoblastic carcinosarcoma, however, both the epithelial and mesenchymal components undergo malignant transformation [13,15,16,21,22,23,24]. While this should make the diagnosis more straightforward, these are most likely also reported as basal cell carcinoma of the sarcomatoid type [13,15,16,21,22,23,24]. To date, there have been only around a dozen cases of true trichoblastic carcinosarcoma reported [13,15,16,21,22,23,24].

In the present case report, to the best of the authors’ knowledge, we report the first depiction of a trichoblastic carcinosarcoma with a prominent, but benign, dendritic melanocyte component—melanotrichoblastic carcinosarcoma.

## 2. Case Presentation

An outpatient, formalin-fixed biopsy specimen was presented to our department by the relatives of an 86-year-old female patient. The patient had multiple cardiovascular comorbidities and was bedbound, with the biopsy performed by a surgeon during a home visit. The specimen was excised from the right arm of the patient. The lesion had initially presented as a small subcutaneous nodule that had been slowly growing for some time until it penetrated the epidermis, at which point it started growing rapidly, increasing in size from 2 cm to nearly 5 cm in the span of three months (Figure 1). The relatives, despite the condition of the patient, elected for tumor excision due to the ulcerated nature of the lesion, continuous bleeding from the lesion and the development of a foul smell from it, indicating infection.

The specimen presented to us was a tumor nodule with an overall size of 4 × 3 × 3 cm after fixation. On cross-section, the specimen was predominantly soft and grayish; however, some areas were more nodular and firm, as well as some being black in color (Figure 2).

Histology of the lesion showed a multiphase tumor, with unique separate components. The first component was represented by large tumor-cell nests with focal peripheral nest palisading, without clefting relative to the surrounding tissues and/or mucin production, focal retiform areas and solid tumor-cell strands of epithelial cells. The epithelial cells had marked atypia—they were nuclearly dominated with prominent anisocytosis and anisokaryotic nuclei with marginated hyperchromatic chromatin and central prominent nucleoli. Some of the nests showed single bi- and multinucleated cells. In contrast, others had zones with an onion-like arrangement of the cells, more abundant cytoplasms with pronounced eosinophilia, and transitioned into keratinized ghost-like cell aggregates and focal primitive hair follicle formation. In other nests, there was central comedo-like necrosis. Mitotic activity was pronounced in the epithelial nests with dominance of atypical forms, mainly triploid, with some hotspots reaching 10 mitoses per single field with a magnification of 400× (Figure 3, Figure 4 and Figure 5). The second component was represented by a tumor stroma, which was hypercellular and eosinophilic with enveloping of the epithelial nests and focal formation of papillary mesenchymal-like bodies and intersecting mesenchymal inclusions. In some areas of the stroma, there were foci of myxoidization with marked enlargement and fusiform stromal cells with ovoid, enlarged and hyperchromic nuclei (low-grade fibromyxosarcoma-like areas), while other areas showed stromal cell multinucleation (Figure 3 and Figure 5). There were no areas within the tumor suggestive of epithelial–mesenchymal transition. The third component was present peripherally in part of the epithelial part of the tumor nests and represented by dendritic cell melanocytes (Figure 4). Tumor-cell emboli were noted within the stromal blood vessels (Figure 5). The initial impression of the tumor based on the described morphological features was that it was a high-grade mixed malignant tumor with probable adnexal differentiation.

Immunohistochemistry was performed with BerEp4 to distinguish the components and differentiate the lesion from a basal cell carcinoma, and showed a typical membrane reaction in the epithelial cells of the nests, but was negative in the stroma (Figure 6). Staining with HMB-45 was performed to prove the melanocyte component and underline the melanocyte morphology, which was dendritic (Figure 6).

Staining with CK20 was performed as most trichoblastic lesions have a small Merkel cell component, and again to differentiate from basal cell carcinoma, in which such components are typically absent, with some of the nests showing a small component of cells with a dot-like reaction (Figure 7). The proliferative index of Ki-67 was high, with some of the nests having nearly 70% positive cells, and stromal zones showed an index of 30% (Figure 7).

Based on the morphological characteristics and immunohistochemical staining patterns, the tumor was interpreted as a malignant tumor of hair follicle origin—high-grade melanotrichoblastic carcinosarcoma, G3 and staged as pT2. As seen in Figure 6, the geographical distribution of the benign dendritic melanocytes indicated that the lesion might have originated from a melanotrichoblastoma that underwent malignant transformation of the epithelial and mesenchymal component, but not of the melanocytic one.

Referral to the oncological committee was suggested to the relatives for further excision, as the specimen did not allow for resection margin interpretation, as well as local radiotherapy. However, due to the condition of the patient, the relatives declined and continued end-of-life care at home.

## 3. Discussion

Trichoblastomas, even in their benign forms, as already mentioned, pose a histopathological difficulty as they are rare and often difficult to differentiate from the much more common basal cell carcinoma, which also develops in the same areas and age groups [2]. Further difficulties arise with some of the exceedingly rare and currently underrecognized skin tumors, as classified by the WHO, such as melanotrichoblastoma [4,5,6,7,8,9,10,11,12]. In melanotrichoblastomas, the differential diagnosis again includes basal cell carcinoma in its pigmented variant, but is also expanded to include melanoma and other exceedingly rare adnexal tumors, such as melanotic matricoma [3,4,5,6,7,8,9,10,11,12,25]. Furthermore, a similar, albeit rare, malignant entry, depicted as a basomelanocytic tumor, which in its nature represents a collision of basal cell carcinoma and melanoma, can exhibit significant morphological overlap [26].

Trichoblastic carcinomas are even rarer than benign trichoblastomas. Once again, this rarity may not accurately reflect the true incidence of the malignancy. However, misidentification with basal cell carcinoma, as the stromal component, which remains benign, is often overlooked due to the tumor’s striking nested appearance with peripheral palisading. A broad discussion can be raised in this context, as some of the identified cases of trichoiblastic carcinomas are aggressive in nature and capable of producing metastasis, whereas historically and conventionally, basal cell carcinoma is known to produce metastasis in exceedingly rare instances [17,18,19,20,27].

Despite fewer than a dozen cases of trichoblastic carcinosarcoma being reported, they have a strictly defined nomenclature with both the epithelial and stromal components undergoing malignant transformation, and based on the degree of atypia and especially the presence of comedo-type necrosis and the degree of stromal anaplasia, they are separated into high- and low-grade [15,16,21,22,23,24]. By definition, in trichoblastic carcinosarcomas, both the epithelial and mesenchymal (stromal) components undergo malignant transformation. As such, they represent a rare group of true carcinosarcomas, with separate epithelial and mesenchymal malignant components and are not tumors wherein the epithelial component undergoes epithelial–mesenchymal transition [22]. Differential diagnosis is again broad and once again includes basal cell carcinoma in its sarcomatoid type; however, in trichoblastic carcinosarcoma, as already mentioned, the epithelial and mesenchymal components are sharply demarcated and do not mix, while sarcomatoid basal cell carcinoma shows evident epithelial–mesenchymal transition [22].

To the best of our knowledge, this is the first reported case of a high-grade trichoblastic carcinosarcoma with a prominent, while still benign, melanocytic component. In our case, both the morphological findings and the immunohistochemical positivity were striking and somewhat definitive. While undoubtedly malignant and ulcerated tumors with extensive pigmentation, especially in an elderly patient, are thought to be melanoma, the nested component had prominent areas of keratinization, akin to keratin pearls in some areas, which immediately poses a counter theory and suggests an epithelial origin. On the other hand, the striking appearance of the stroma, which, although intimately connected with the nested component, remained separate and did not show a gradient towards the nests, posed further questions, as it can be observed both in the sacomatous variant of carcinomas and in melanoma. As such, immunohistochemistry was vital in this case. BerEp4 underlined the epithelial component of the tumors, and together with the staining pattern, helped differentiate from basal cell carcinoma and melanocytic matricoma, although additional future studies are needed to validate BerEp4 staining patterns in such lesions [3,4,28]. HMB-45, while not the most specific melanocyte marker, underlines not only the focal presence of melanocytes within the nests, differentiating them from artificial pigmentation such as hemosiderin pigment, but also highlights the physiological dendritic cell morphology of the melanocytes, helping to differentiate them from melanoma [8]. The focal presence of dot-like CK20 positivity in some cells helped identify them as Merkel cells, which have been shown to be a regular component of trichoblastomas and their derivatives and are exceedingly rare in other malignancies [2,3]. Despite the significant mitotic activity that was present with the epithelial component, Ki-67 was also helpful, predominantly in distinguishing the stromal component as malignant, with a high index, which would not be the case in significant degenerative-type atypia in the stroma, which could mimic the morphological findings depicted [13,15,16,21,22,23,24].

While the dendritic melanocyte component remains benign in our case, as already mentioned, the geographical distribution of melanocytes within the nest is indicative of the malignant transformation from a previous melanotrichoblastoma [4,5,6,7,8,9,10,11,12].

### Limitations

The depiction of a single case is an evident limiting factor of the presented report. While both morphological and immunohistochemical depictions of the individual components are present, the main drawback of the presented case is the lack of molecular data regarding the tumor and its individual components. As for data, there have been very few cases of benign melanotrichoblastomas and malignant trichoblastic carcinosarcomas; future studies are encouraged to include such data or collect samples from already published cases, as a large-scale study on such exotic entries seems unlikely. Furthermore, the authors hope the presented case will raise awareness for these rare tumors and help in their morphological and immunohistochemical misdiagnosis so as to avoid them being designated as other malignancies.

## 4. Conclusions

Trichoblastomas are rare and underrecognized tumors. They have several variants with an exceedingly rare occurrence, such as melanotrichoblastomas, and can undergo malignant transformation into both trichoblastic carcinomas and trichoblastic carcinosarcomas. To the best of the authors’ knowledge, the presented case is the first reported one of a high-grade melanotrichoblastic carcinosarcoma. While neither melanotrichoblastomas nor the currently proposed terminology of melanotrichoblastic carcinosarcoma is accepted by the WHO, the authors hope that future studies and revisions of the classification will include these exotic entries, further raising awareness of them.

## Figures and Tables

**Figure 1 reports-08-00218-f001:**
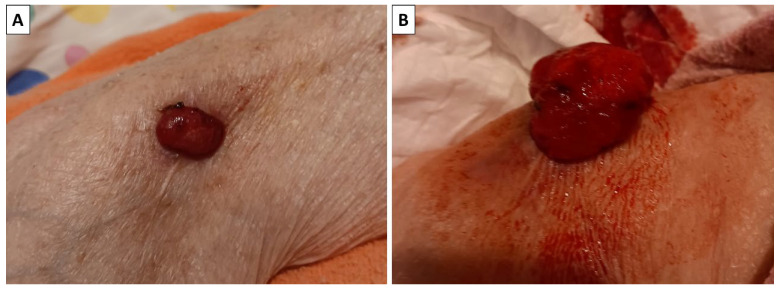
Clinical progression of the lesion. (**A**) The lesion when it initially perforated the skin; (**B**) the lesion three months later.

**Figure 2 reports-08-00218-f002:**
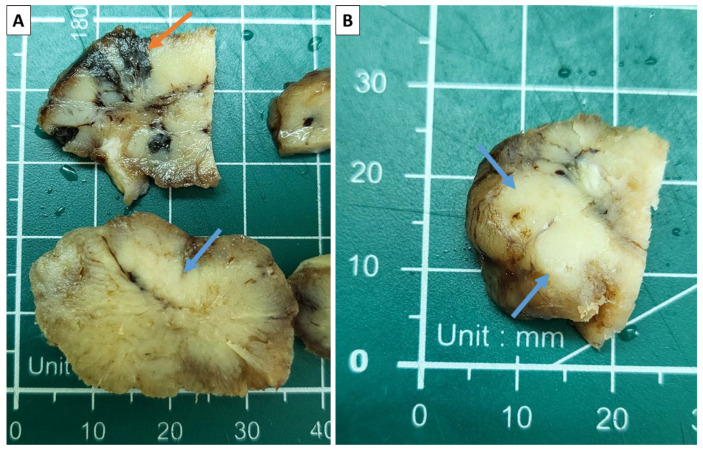
Sectioning of the specimen. (**A**) Areas of intense pigmentation (orange arrow) and more firm grayish zones (blue arrows); (**B**) solid, nested zones appearing on the section (blue arrows).

**Figure 3 reports-08-00218-f003:**
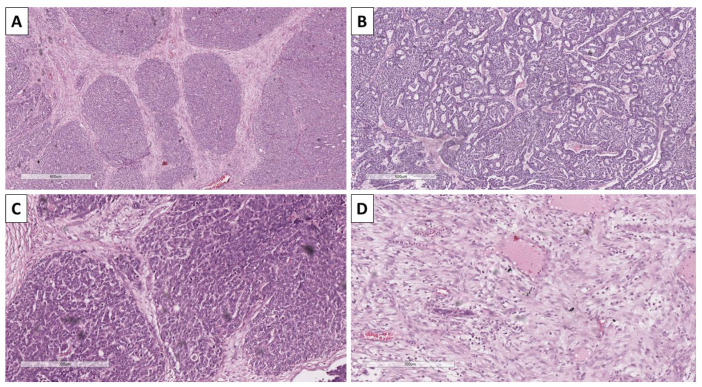
Histopathology of tumor components. (**A**) Nests with peripheral palisading, H&E stain, original magnification 100×; (**B**) retiform areas, H&E stain, original magnification 100×; (**C**) stroma with focal intersection of the nest and formation of papillary mesenchymal-like bodies, H&E stain, original magnification 100×; (**D**) bizarre stromal component, H&E stain, original magnification 100×.

**Figure 4 reports-08-00218-f004:**
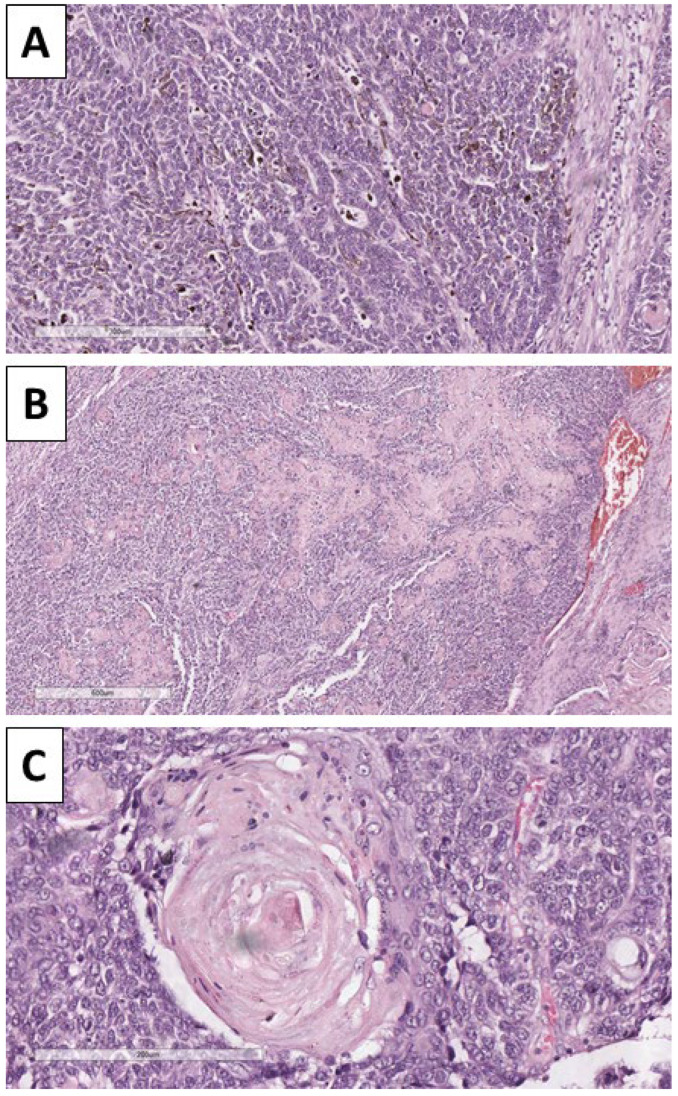
Histopathology of the tumor. (**A**) Dendritic melanocytes within the nest, H&E stain, original magnification 100×; (**B**) areas of keratinization, H&E stain, original magnification 40×; (**C**): areas of keratin pearl-like formation with ghost-like cells, H&E stain, original magnification 200×.

**Figure 5 reports-08-00218-f005:**
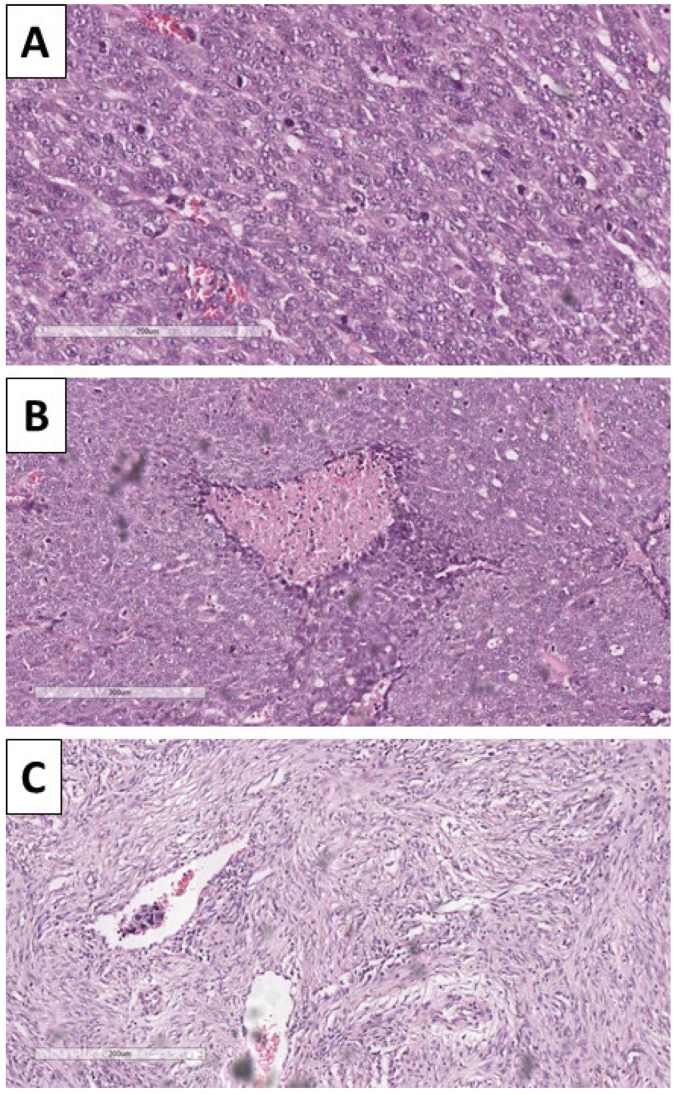
Histopathology of the tumor. (**A**) Pronounced mitotic activity, H&E stain, original magnification 200×; (**B**) areas of comedo-like necrosis centrally within the nests, H&E stain, original magnification 100×; (**C**) bizarre stroma with tumor-cell emboli in the blood vessels, H&E stain, original magnification 100×.

**Figure 6 reports-08-00218-f006:**
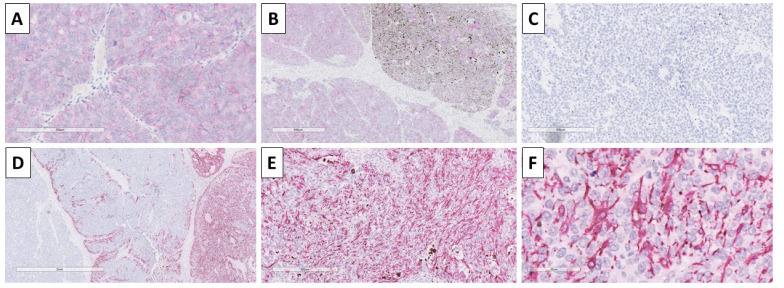
Immunohistochemistry of the tumor. (**A**) BerEp4 positivity in tumor nest, original magnification 20×; (**B**) BerEp4 positivity in pure epithelial and mixed epithelial and melanocyte-rich nests, negative in the stroma, original magnification 40×; (**C**) HMB-45 negative in pure epithelial nests, original magnification 200×; (**D**) HMB-45 negative in stroma and pure epithelial nests, positive in mixed epithelial and melanocyte rich nests, original magnification 40×; (**E**) HMB-45 positive in melanocyte rich nests, original magnification 100×; (**F**) HMB-45 underlining the dendritic cell morphology of melanocytes, original magnification 400×.

**Figure 7 reports-08-00218-f007:**
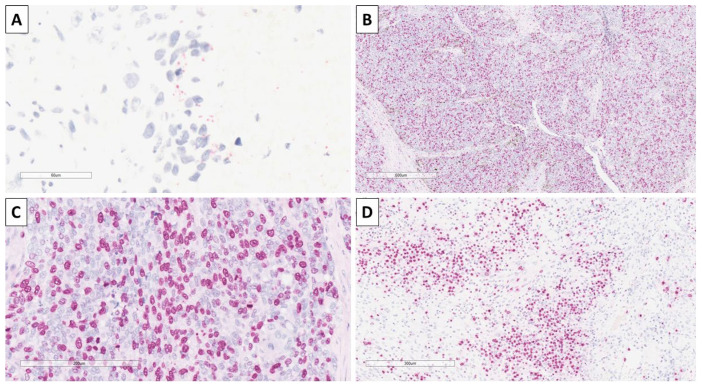
Immunohistochemistry of the tumor. (**A**) Focal CK20 dot-like reaction, original magnification 400×; (**B**) high Ki-67 proliferative index in epithelial component, original magnification 40×; (**C**) high Ki-67 proliferative index in epithelial component, original magnification 200×; (**D**) high Ki-67 proliferative index in stromal component, original magnification 100×.

## Data Availability

The original contributions presented in this study are included in the article. Further inquiries can be directed to the corresponding author.

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
