# Peer review of "Melanotrichoblastic Carcinosarcoma: A Histopathological Case Report of a Previously Undescribed Nosological Unit"

_reports, 2025, doi:10.3390/reports8040218_

Round 1
Reviewer 1 Report
Comments and Suggestions for Authors
In this case report, the authors present a case of melanotrichoblastic carcinosarcoma. Given that this is an exceptionally rare pathological diagnosis, it is imperative that the supporting evidence be presented with great care and in a stepwise manner. My specific comments are as follows:
1: Clinical findings: The tumor appears red, with only minimal dark areas suggestive of melanin. Histologically, some tumor cells appear to contain melanin pigment. In this context, the authors should clarify whether it is justified to designate the tumor as melanotrichoblastic, and provide a clear rationale for this terminology.
2: In Figure 5C, the so-called “bizarre stroma” is shown. The authors should clarify whether this component represents tumor cells or stroma. If tumor cells are indeed present, please indicate explicitly which cells are neoplastic (e.g., by arrows or other markers).
3: The tumor is described as a carcinosarcoma. Does the “sarcoma” component correspond to the stromal-like area mentioned in comment 2? It is critical to determine whether the stromal component truly consists of neoplastic cells. The authors should provide convincing evidence of a transition from the epithelial component to the mesenchymal component. Furthermore, immunohistochemical data such as E-cadherin, N-cadherin, and BerEP4 expression in these transition areas would be highly informative. Without such evidence, the diagnosis of sarcoma in this case remains uncertain.
Author Response
In this case report, the authors present a case of melanotrichoblastic carcinosarcoma. Given that this is an exceptionally rare pathological diagnosis, it is imperative that the supporting evidence be presented with great care and in a stepwise manner. My specific comments are as follows:
1: Clinical findings: The tumor appears red, with only minimal dark areas suggestive of melanin. Histologically, some tumor cells appear to contain melanin pigment. In this context, the authors should clarify whether it is justified to designate the tumor as melanotrichoblastic, and provide a clear rationale for this terminology.
- Dear reviewer, thank you this comment. Grossly the tumor appears red, however, after fixation and on cross section, as seen in the figures, it has extensive pigmented areas. As seen in multiple places within the text, not only the cell morphology of melanocytes is described, but also IHC was performed to show the distribution of these melanocytes (Figure 4A and 6 C-F). Hence the presence of melanocytes within the tumor is unquestionable, justifying the terminology proposed.
2: In Figure 5C, the so-called “bizarre stroma” is shown. The authors should clarify whether this component represents tumor cells or stroma. If tumor cells are indeed present, please indicate explicitly which cells are neoplastic (e.g., by arrows or other markers).
- Dear reviewer, thank you this comment. The stromal component is a secondary special cell population within all trichoblastomas. The bizarre cells depicted in the figure are the malignant transformation of the stroma. The whole figure shows this population, hence adding arrows would suggest that only some of the cells have undergone malignant transformation.
3: The tumor is described as a carcinosarcoma. Does the “sarcoma” component correspond to the stromal-like area mentioned in comment 2? It is critical to determine whether the stromal component truly consists of neoplastic cells. The authors should provide convincing evidence of a transition from the epithelial component to the mesenchymal component. Furthermore, immunohistochemical data such as E-cadherin, N-cadherin, and BerEP4 expression in these transition areas would be highly informative. Without such evidence, the diagnosis of sarcoma in this case remains uncertain.
- Dear reviewer, thank you this comment. Trichoblastomas are mixed tumors that have two separate cell population – epithelial and mesenchymal, akin to fibroadenoma of the breast and pleomorphic adenoma of the salivary glands. Hence, the term carcinosarcoma refers not to an epithelial tumor that has undergone mesenchymal transition, but a true mixed tumor wherein both components have undergone malignant transformation at the same time. As such no evidence can be produced for the transformation as this is not the case in the resent report. Regarding the request for IHC, all suggested molecules are epithelial surface proteins and will be negative in the malignant stromal component of the tumor. Furthermore, BerEP4 is already presented in the text and shows positivity only in the epithelial component (Figure 6 A and B).
Reviewer 2 Report
Comments and Suggestions for Authors
This case report provides a valuable and well-documented description of a very rare tumor.
The abstract is concise but would benefit from emphasizing the clinical impact and diagnostic challenges more explicitly.
The introduction could be streamlined to avoid redundancy, particularly regarding trichoblastoma–basal cell carcinoma overlap.
The case report itself is very detailed and beautifully described. Clinical history would be clearer with more structured presentation (timeline, comorbidities, indication for biopsy).
The discussion is thorough but sometimes reiterates background information rather than focusing on the unique aspects of this case and its implications for future classification and diagnostic practice. Please focus on the case more in this section.
Limitations should mention the lack of molecular analysis and the absence of follow-up due to patient s condition.
Conclusions are valid but could be framed more cautiously to reflect that this is a single case and that additional reports are necessary before defining a new nosological entity.
Author Response
This case report provides a valuable and well-documented description of a very rare tumor.
- Dear reviewer, thank you for this comment.
The abstract is concise but would benefit from emphasizing the clinical impact and diagnostic challenges more explicitly.
- Dear reviewer, thank you for this comment. Changes will be made in accordance with this comment.
The introduction could be streamlined to avoid redundancy, particularly regarding trichoblastoma–basal cell carcinoma overlap.
- Dear reviewer, thank you for this comment. Changes will be made in accordance with this comment.
The case report itself is very detailed and beautifully described. Clinical history would be clearer with more structured presentation (timeline, comorbidities, indication for biopsy).
- Dear reviewer, thank you for this comment. Detailed clinical history, while we agree is important, would mostly distract from the morphological findings, as the patient has had several myocardial and cerebral infarctions, as well as long lasting congestive heart failure, which are not connected to the process of oncogenesis. Indication for resection is also mentioned within the text - The relatives, despite the condition of the patient, elected for tumor excision due to the ul-cerated nature of the lesion, continuous bleeding from the lesion and the development of a foul smell from it, indicating infection.
The discussion is thorough but sometimes reiterates background information rather than focusing on the unique aspects of this case and its implications for future classification and diagnostic practice. Please focus on the case more in this section.
- Dear reviewer, thank you this comment. Changes will be made in accordance with this comment.
Limitations should mention the lack of molecular analysis and the absence of follow-up due to the patient's condition.
- Dear reviewer, thank you for this comment. Changes will be made in accordance with this comment.
Conclusions are valid but could be framed more cautiously to reflect that this is a single case and that additional reports are necessary before defining a new nosological entity.
- Dear reviewer, thank you for this comment. Changes will be made in accordance with this comment.
Reviewer 3 Report
Comments and Suggestions for Authors
Trichoblastomas and their variants are rare and often underrecognized skin tumors. They are frequently misdiagnosed as basal cell carcinoma due to similar appearances. Even rarer variants include melanotrichoblastoma and malignant forms like trichoblastic carcinoma and trichoblastic carcinosarcoma, which are likely even more overlooked. In the manuscript “Melanotrichoblastic Carcinosarcoma: A Histopathological Case Report of A Previously Undescribed Nosological Unit”, Stoyanov and Popov have shown that an 86-year-old woman had a tumor on her right arm. Histological examination showed a mix of malignant epithelial nests, retiform structures, focal keratinisation, comedo-type necrosis, dendritic melanocytes, and an abnormal, hypercellular stroma. Immunohistochemistry findings demonstrate BerEp4 positive in epithelial nests, HMB-45 positive in dendritic melanocytes, CK20 focal positivity in Merkel cells and stroma negative for epithelial and melanocytic markers but showed high Ki-67, indicating malignancy. Based on these features, the tumor was diagnosed as a melanotrichoblastic carcinosarcoma. This case report represents the first known report of melanotrichoblastic carcinosarcoma, highlighting the need for greater awareness and diagnostic recognition of these extremely rare tumor types.
Minor comments:
- Use of HMB-45 as the marker for the melanoma staining: The authors have used the HMB-45 as the marker of melanoma but in order to increase the sensitivity of the staining I would suggest the authors to use the cocktail of MART-1, s100 and HMB-45 which will also distinguish the early pre-melanoma cells with utmost sensitivity PMID: 18399807.
Author Response
Trichoblastomas and their variants are rare and often underrecognized skin tumors. They are frequently misdiagnosed as basal cell carcinoma due to similar appearances. Even rarer variants include melanotrichoblastoma and malignant forms like trichoblastic carcinoma and trichoblastic carcinosarcoma, which are likely even more overlooked. In the manuscript “Melanotrichoblastic Carcinosarcoma: A Histopathological Case Report of A Previously Undescribed Nosological Unit”, Stoyanov and Popov have shown that an 86-year-old woman had a tumor on her right arm. Histological examination showed a mix of malignant epithelial nests, retiform structures, focal keratinisation, comedo-type necrosis, dendritic melanocytes, and an abnormal, hypercellular stroma. Immunohistochemistry findings demonstrate BerEp4 positive in epithelial nests, HMB-45 positive in dendritic melanocytes, CK20 focal positivity in Merkel cells and stroma negative for epithelial and melanocytic markers but showed high Ki-67, indicating malignancy. Based on these features, the tumor was diagnosed as a melanotrichoblastic carcinosarcoma. This case report represents the first known report of melanotrichoblastic carcinosarcoma, highlighting the need for greater awareness and diagnostic recognition of these extremely rare tumor types.
- Dear reviewer, thank you this comment.
Minor comments:
- Use of HMB-45 as the marker for the melanoma staining: The authors have used the HMB-45 as the marker of melanoma but in order to increase the sensitivity of the staining I would suggest the authors to use the cocktail of MART-1, s100 and HMB-45 which will also distinguish the early pre-melanoma cells with utmost sensitivity PMID: 18399807.
- Dear reviewer, thank you for this comment. Multiple instances within the text underscore that the melanocytic component within the tumor is completely benign. HMB-45 was performed with the goal of better underlining this component, not as evidence for malignant transformation, which is not present in the case (for this cellular component of the tumor).
Round 2
Reviewer 1 Report
Comments and Suggestions for Authors
Thank you for your reply. Since the cells observed in the stroma are also considered malignant, I would appreciate it if you could include the contents of your reply in the Discussion or other relevant sections of the manuscript. It would also make the report more convincing if you could show histopathological images demonstrating continuity or transitional areas between the main tumor and the stromal component. Thank you for giving me the opportunity to review your manuscript. I have no further comments.
Author Response
Thank you for your reply. Since the cells observed in the stroma are also considered malignant, I would appreciate it if you could include the contents of your reply in the Discussion or other relevant sections of the manuscript. It would also make the report more convincing if you could show histopathological images demonstrating continuity or transitional areas between the main tumor and the stromal component. Thank you for giving me the opportunity to review your manuscript. I have no further comments.
- Dear reviewer, thank you for these comments. As already mentioned, the stromal malignant component is entirely separate from the epithelial one; we have expanded relevant sections of the manuscript to further stress this, as suggested. Due to the separate nature of the malignant components, as explained, we cannot produce a figure showing epithelial-mesenchymal transition, as such a phenomenon is not present in the case (true carcinosarcoma).